Melatonin attenuates MPP+-induced autophagy via heat shock protein in the Parkinson’s disease mouse model

Guo Yinli 1
Liu Chengbo 2 liuchengbo201509@163.com
1 Institute of Innovation and Entrepreneurship, Guizhou Education University , Guizhou, Wudang District, Guiyang City , China
2 Medical section, Jiang Ling County People’s Hospital , Hubei, Jiangling County, Jingzhou City , China
Leppik Liudmila
Electronic publication date: 2025 Jan 21
Publication date: 2025
Volume: 13
Electronic Location ID: e18788
Received 2024 Sep 10; Accepted 2024 Dec 10
Copyright: © 2025 Guo and Liu
Copyright year: 2025
Copyright holder: Guo and Liu
License: This is an open access article distributed under the terms of the Creative Commons Attribution License, which permits unrestricted use, distribution, reproduction and adaptation in any medium and for any purpose provided that it is properly attributed. For attribution, the original author(s), title, publication source (PeerJ) and either DOI or URL of the article must be cited.
License URL: https://creativecommons.org/licenses/by/4.0/

Keywords: Parkinson’s disease, Melatonin, HSP70, Autophagy, CDK5, LC3 II /LC3I, p62

Funding: 2023 General Undergraduate College Scientific Research Project (Youth Project) of Guizhou Education Department QJJ-2022246 This research was funded by the 2023 General Undergraduate College Scientific Research Project (Youth Project) of Guizhou Education Department [QJJ-2022246]. The funders had no role in study design, data collection and analysis, decision to publish, or preparation of the manuscript.

==============================
Background

This study investigates the protective properties of melatonin in an in vivo Parkinson’s disease (PD) model, focusing on the underlying mechanisms involving heat shock proteins (HSPs).

Methods

Twelve adult male C57BL/6 mice were randomly divided into four groups (normal control, melatonin control, Parkinson’s model, and melatonin treatment; n = 3 per group) and housed in a single cage. 1-methyl-4-phenyl-1,2,3,6-tetrahydropyridine (MPTP) was injected intraperitoneally in the Parkinson’s model and treatment groups to establish a subacute PD model, while controls received saline. Limb motor ability was assessed 1 h after the final injection using behavioral tests, including the open field test to evaluate central zone entries and average movement. Dopamine transporter (DAT) expression in the striatum was analyzed by immunohistochemistry, and Western blot was used to measure autophagy proteins and HSP70 levels.

Results

The PD mouse model was successfully established through MPTP stimulation. Compared to the normal control group, the model group showed a significant reduction in the frequency of entering the central zone and average movement. The number of DAT-positive cells in the brain also decreased significantly. The expression levels of HSP70 and CDK5 were significantly lower, while the expression levels of LC3 II /LC3I and p62 increased significantly. In the MT treatment group, both the frequency of entering the central zone and the average movement were significantly higher compared to the model group. DAT-positive cells in the midbrain also increased significantly. The expression levels of HSP70 and CDK5 were significantly elevated, while the expression levels of LC3 II /LC3I and p62 protein were significantly decreased.

Conclusion

Melatonin exerts a protective effect against MPP+-induced damage to dopaminergic neurons, presumably by upregulating HSP70, which inhibits neuronal autophagy.

Introduction

Parkinson’s disease (PD) is the second most prevalent neurodegenerative disorder, primarily affecting middle-aged and elderly populations, with a multifactorial etiology involving genetic, environmental, and neuroimmune factor (Lauritsen & Romero-Ramos, 2023). The hallmark of PD pathology is the progressive degeneration of dopaminergic (DA) neurons in the substantia nigra, accompanied by the formation of Lewy bodies composed primarily of α-synuclein (SNCA) (Burré, Sharma & Südhof, 2018; L’Episcopo et al., 2018; Raza, Anjum & Shakeel, 2019). Clinically, PD manifests with motor symptoms such as tremors, bradykinesia, and gait instability, as well as non-motor symptoms like olfactory decline, sleep disorders (Bjørklund et al., 2020; Bucur & Papagno, 2023), and gastrointestinal dysfunction. With an aging global population, the prevalence of PD is projected to rise significantly, affecting up to 12.9 million people by 2040, thereby imposing a substantial socioeconomic burden (Bloem, Okun & Klein, 2021; Dorsey & Bloem, 2018). Despite its growing impact, the precise pathogenesis of PD remains unclear (Emamzadeh & Surguchov, 2018), limiting the development of effective therapeutic strategies.

Emerging evidence suggests that autophagy, a key cellular process for maintaining homeostasis through the degradation of damaged organelles and proteins, plays a pivotal role in PD (Liu et al., 2023; Rickman, Hilyard & Heckmann, 2022). Autophagy dysfunction has been linked to the accumulation of SNCA and other toxic aggregates in dopaminergic neurons (Nechushtai, Frenkel & Pinkas-Kramarski, 2023; Simon, Tanner & Brundin, 2020; Wang et al., 2020). Markers such as the LC3-II/LC3-I ratio and p62 protein levels are commonly used to evaluate autophagy activity (Mizushima & Komatsu, 2011). Cyclin-dependent kinase 5 (CDK5), a neuronal kinase implicated in basal autophagy regulation, has been shown to modulate neurodegeneration (Nandi et al., 2017). Meanwhile, heat shock protein 70 (HSP70), a stress-induced molecular chaperone, is known to exert neuroprotective effects under oxidative stress conditions but can also suppress autophagy when overexpressed (Witkin, Kanninen & Sisti, 2017). Understanding the intricate interplay between these factors is crucial for elucidating PD pathogenesis and exploring potential therapeutic interventions.

The neurotoxin 1-methyl-4-phenyl-1,2,3,6-tetrahydropyridine (MPTP) is widely used to create experimental PD models due to its selective toxicity to dopaminergic neurons in the substantia nigra (Hare et al., 2013). This model closely mimics many pathological and behavioral features of PD, making it a valuable tool for studying disease mechanisms and testing potential treatments. In this study, the MPTP mouse model was employed to investigate the effects of melatonin (MT) on autophagy and dopaminergic neuron survival.

Melatonin, a neurohormone produced by the pineal gland, exhibits diverse biological functions, including antioxidative, anti-inflammatory, and neuroprotective effects (Li et al., 2017). Previous research has shown that melatonin attenuates neurodegeneration in PD by mitigating oxidative stress, reducing SNCA aggregation, and modulating autophagy (Jung, Choi & Oh, 2022; Pandi-Perumal et al., 2013). Specifically, melatonin has been reported to downregulate CDK5-mediated autophagy and improve mitochondrial quality control (Chuang et al., 2016; Su et al., 2015). However, its impact on the CDK5/LC3/p62 axis and its interplay with HSP70 in PD models remains inadequately explored.

This study aims to evaluate the neuroprotective effects of melatonin in an MPTP-induced PD mouse model, focusing on the regulation of autophagy via the CDK5/LC3/p62 axis and the role of HSP70 in this process. By investigating these pathways, this work seeks to provide new insights into the mechanisms underlying PD pathogenesis and to explore the therapeutic potential of melatonin as a neuroprotective agent.

Materials and Methods

Animal grouping and modeling

Twelve SPF C57BL/6 male mice (8–10 weeks, 25–30 g) purchased from Sibeifu Biotechnology Co., Ltd., Beijing (China). Mice were maintained and bred in specific pathogen-free conditions at the Tianjin Jinke Bona Biotechnology Co., Ltd., whose animal Ethics Committee provided full approval for this research (approval number: GENINK-20230036). The mice were housed three per cage under a 12/12-h light/dark cycle at 22–23 °C with 40–70% humidity. They had free access to food and water and were acclimated for 1 week before the experiment.

They were equally divided into four groups: normal control, melatonin control, Parkinson’s model and melatonin treatment, with three mice in each group. The PD model was induced by intraperitoneal injection of MPTP (20 mg/kg) twice daily, with melatonin (10 mg/kg) administered half an hour after each MPTP injection for seven consecutive days. One hour after the last dose, the limb motor ability of the mice was measured by behavioral method. Mice was euthanized by carbon dioxide immediately followed by cervical dislocation for further ex vivo analyses. After euthanasia, the brain were harvested and cleaned with ice-cold PBS. Each mouse brain tissue were cut in half, one half fixed at 4 °C with 4% paraformaldehyde (PFA, pH 7.4) for 48 h and then embedded in paraffin for histological analysis and the remaining half snap-frozen in liquid nitrogen before stored at −80 °C for Western blot analysis.

Behavior tests

Behavioral tests began 1 day after the final drug injection following adaptive training. The open field test (OFT) was used to evaluate mice’s autonomy and exploratory behavior in response to a novel environment, reflecting anxiety levels. Mice were placed in the center of an open field box (50 × 50 × 40 cm3), divided into 16 compartments (12 peripheral and four central). They were allowed to move freely for 1 min while the VisuTrack video analysis system recorded immobility time in total and non-central regions, as well as the number and duration of entries into the central area. The box was cleaned with 75% alcohol after each experiment to remove odors. Reduced entries into the central zone, lower moving speed, and decreased total distance traveled indicated heightened anxiety.

Western blotting (WB)

The brain tissues stored at −80 °C were taken out and homogenized in a cold lysis buffer containing protease and phosphatase inhibitors (Cat# R0020; Solarbio, Beijing, China) and centrifuged at 12,000× g for 15 min at 4 °C. The supernatant was transferred to a fresh tube and measured for protein concentration using BCA assay (Cat# P0011; Beyotime, Shanghai, China). The lysate was diluted to a uniform protein concentration with lysis.

The samples were prepared for western blot analysis by mixing the diluted lysate with 5× protein loading buffer (Cat# P0015L; Beyotime, Shanghai, China), and then boiling at 100 °C for 10 min. The boiled samples were centrifuged briefly and 20 µg of each sample was taken to load into the wells of the 10% SDS-PAGE gel, along with a molecular weight marker (Cat# P1200; Solarbio, Beijing, China) for electrophoresis.

Separated proteins were transferred from SDS-PAGE gels to PVDF membranes using the wet transfer method at 100V for 1 h (Spencer et al., 2000). The membranes were then blocked in Tris-buffered saline solution with 0.1% Tween-20 (TBST) and 5% skim milk. After washing three times with TBST, the membranes were incubated with primary antibodies against HSP70 (1:1,000, CST #4872), CDK5 (1:1,000, CST #2506), LC3 II /LC3I (1:1,000, CST #4108), p62 (1:1,000, CST #5114) and GAPDH (UM4002; Tianjin Youkang Biotech, Tianjin, China, diluted to 1:2,000) for 1 h at room temperature. After washing three times with TBST, the membranes were incubated with horseradish peroxidase (HRP)-labeled secondary antibodies for 0.5 h at room temperature. Finally, the immunoblots were visualized using the chemiluminescence method (Tanon 5200; Shanghai Tianneng, China). The relative protein levels were measured by Gel-Pro analyzer (Media Cybernetics, Rockville, MD, USA).

Immunohistochemistry

The PFA fixed tissues were washed with PBS and then went through dehydration by immersing in a graded ethanol series (70% ethanol, 1 h; 80% ethanol, 1 h; 95% ethanol, 1 h; and 100% ethanol, 2 h), clearing with xylene at room temperature for 2 h, infiltration with melted paraffin wax at 60 °C for 2 h. Treated brain tissues were put into a mold filled with melted paraffin and oriented appropriately to allow the paraffin to solidify at room temperature. After that, the embedded tissue blocks were stored at 4 °C until sectioning. Paraffin sections of brain tissue (4 μm thick) were stained with DAT antibody (1:400, cat#ab184451, Abcam, Cambridge, MA, USA) and incubated with horseradish peroxidase (HRP) for 50 min. The target protein was stained by the DAB staining method. Staining results were examined using an optical microscope (DP26, OLYMPUS, Tokyo, Japan). For analysis, at least six fields of view at 20× magnification were randomly selected from each section. DAT expression in brain tissue was quantitatively assessed using Image-Pro Plus 6.0 software (Media Cybernetics, Inc., Rockville, MD, USA) by calculating the integrated optical density (IOD) per region of interest (AOI) in the positively stained areas.

Statistical analysis

All data were analyzed by either SPSS version 22.0 (IBM Corp, Armonk, New York, USA) or GraphPad Prism (GraphPad Software, San Diego, CA, USA). All values are expressed as mean ± SD. Unpaired Student’s two-tailed t test was used for two sample data comparison. Two-way ANOVA was used for multiple comparisons. The correlations were determined by calculating Pearson linear correlation coefficients or Spearman rank correlation coefficients. Significance levels were indicated as follows: *p < 0.05, **p < 0.01, ***p < 0.001.

Results

Effects of melatonin on behavior of PD mice

Mouse behavior analysis was performed with the open field test where mouse movement was tracked and evaluated with the VisuTrack tracking system (Fig. 1A). The average movement was significantly reduced in the Parkinson’s model group compared to the control group (t = 5.048, p < 0.05), confirming impaired motor function. Conversely, the treatment group showed a significant improvement in movement compared to the model group (t = 4.311, p < 0.05). Similarly, the number of times the mice entered the central zone was significantly lower in the model group than in the MT control group (t = 7.326, p < 0.05). However, this metric significantly increased in the treatment group compared to the model group (t = 5.500, p < 0.05), further validating the successful establishment of the Parkinson’s disease model (Figs. 1B, 1C).

Figure 1 Effects of melatonin on behavior of PD mice.

(A) Open-field behavior tracking heat map. Representative images depicted the time spent at different locations within the open field box. (B) Effects of melatonin on the movement of PD mice. (C) Effects of melatonin on the number of times entering the central in PD mice. Data are expressed as mean ± SD and statistical significance as *, p < 0.05.

Effects of melatonin on dopamine neurons in the substantia nigra of PD mice

Compared to the control group, DAT-positive nerve fibers in the substantia nigra of the MPTP group and MT treatment group were significantly reduced (t = 3.852, p < 0.01 and t = 2.919, p < 0.05). Melatonin treatment significantly increased the number of DAT-positive nerve fibers (t = 3.443, p < 0.01) (Figs. 2A and 2B).

Figure 2 Effects of melatonin on dopamine neurons in the substantia nigra of PD mice.

(A) Representative IHC of DAT in brain tissue from each group. Scale bar: 50 μm. (B) Quantification of DAT staining. Statistical significance is indicated by *, p < 0.05; **, p < 0.01; or ***, p < 0.001.

Effects of melatonin on HSP70 protein expression in the PD mouse model

HSP70 protein expression in the MPTP group, melatonin group and melatonin treatment group were significantly decreased compared to the control group (t-test showed p < 0.01). Melatonin treatment and melatonin group significantly increased HSP70 protein expression compared to the MPTP group (t-test showed p < 0.05) (Figs. 3A and 3B).

Figure 3 Effects of melatonin on HSP70 protein expression in the PD mouse model.

(A) A representative immunoblotting of HSP70. GADPH was used as an internal control for proteins. (B) The ratio of HSP70. Protein levels were expressed as the mean ± SD (n = 3). Statistical significance is indicated by *, p < 0.05; **, p < 0.01; or ***, p < 0.001.

Effects of melatonin on CDK5, LC3 II /LC3I, and p62 protein expression in the PD mouse model

Western blot analysis showed that the relative protein expressions of CDK5 protein expression was significantly inhibited in the MPTP group compared to the control group (t = 122.0, p < 0.0001), while melatonin treatment significantly increased CDK5 expression (Figs. 4A, 4B). LC3 II/LC3I and p62 in MPTP groups and melatonin treatment group were significantly higher than those in control group (t-test showed p < 0.05). while melatonin treatment significantly decreased it (t = 13.09, p < 0.01) (Figs. 4C, 4D).

Figure 4 Effects of melatonin on CDK5, LC3 II/LC3I, and p62 protein expression in the PD mouse model.

(A) A representative immunoblotting of CDK5, LC3II/LC3I, and p62. GADPH was used as an internal control for proteins. (B–D) The ratio of CDK5, LC3II/LC3I, and p62 respectively. Protein levels are expressed as the mean ± SD (n=3). Statistical significance is indicated by *, p < 0.05; **, p < 0.01; or ***, p < 0.001.

Discussion

PD is a neurodegenerative disorder characterized by the progressive loss of dopaminergic neurons, with mitochondrial dysfunction and disrupted autophagy playing critical roles in its pathogenesis. Melatonin, a neurohormone with antioxidative and anti-inflammatory properties, has shown promise as a neuroprotective agent in PD models (Biswal et al., 2024). This study demonstrated that melatonin significantly mitigates dopaminergic neuronal degeneration in an MPTP-induced PD mouse model, supporting its potential as a therapeutic candidate.

One of the key findings of this study is the upregulation of heat shock protein 70 (HSP70) by melatonin in the PD mouse model. HSP70 is a stress-induced chaperone protein with cytoprotective roles, particularly under conditions of oxidative stress, which are prevalent in PD. Persistent oxidative stress has been reported to suppress HSP70 expression, exacerbating neuronal damage (Li et al., 2019; Yamashima, 2016). By increasing HSP70 levels, melatonin may counteract the damaging effects of oxidative stress, thus protecting neuronal integrity (Rastogi & Haldar, 2020). Additionally, previous studies have revealed that HSP70 negatively regulates autophagy, partly through its effect on AMPK signaling (Alhasan et al., 2024; Song et al., 2023). This aligns with our findings, which indicate that melatonin decreases LC3B-II and p62 protein levels, key markers of autophagy, and inhibits excessive autophagic activity in the PD model.

Melatonin’s dual ability to modulate autophagy and enhance HSP70 expression likely plays a central role in its neuroprotective effects. Autophagy, while essential for cellular homeostasis, can become dysregulated in PD, contributing to neuronal dysfunction. Our results suggest that melatonin fine-tunes autophagy by promoting mitophagy, the selective removal of damaged mitochondria, while suppressing excessive autophagy that could lead to cell death. This is consistent with earlier findings that melatonin supports mitochondrial quality control and mitigates mitochondrial dysfunction in PD models (Biswal et al., 2024). Importantly, these findings highlight a potential therapeutic mechanism whereby melatonin maintains a balance between autophagy and mitophagy to support neuronal survival.

However, this study has several limitations that warrant consideration. First, while the MPTP-induced PD mouse model replicates many aspects of PD pathology, it does not fully capture the complexity of the human disease, particularly the non-motor symptoms and long-term disease progression. Thus, translating these findings to clinical applications requires caution. Second, the small sample size of three mice per group limits the statistical power of our results. Future studies should include larger cohorts to strengthen the reliability and generalizability of the findings. Additionally, only male mice were used in this study. Given the potential influence of sex-specific factors on PD pathogenesis and treatment responses, future research should incorporate female mice to provide a more comprehensive understanding.

Another area requiring further investigation is the precise mechanism through which melatonin modulates autophagy and HSP70 expression. While this study establishes a correlation, the signaling pathways mediating these effects remain unclear. For instance, the interplay between melatonin, AMPK signaling, and HSP70 in the regulation of autophagy deserves closer examination. Moreover, understanding whether melatonin’s effects are dose-dependent and whether prolonged administration could have adverse consequences is critical for its potential therapeutic application.

Conclusion

This study demonstrates that melatonin exerts neuroprotective effects in an MPTP-induced PD mouse model by upregulating HSP70 expression and modulating autophagy. These findings provide preliminary evidence supporting the use of melatonin as a potential treatment strategy for PD. Nevertheless, significant gaps remain in our understanding of its clinical relevance and underlying mechanisms. Future studies should address these limitations by using larger and more diverse sample populations, exploring the molecular pathways involved, and evaluating melatonin’s effects in models that better mimic human PD. By doing so, we can build a stronger foundation for translating these findings into clinical therapies for PD.

Supplemental Information

Supplemental Information 1 Supplemental.

Supplemental Information 2 Author Checklist.

Supplemental Information 3 uncropped gels.

Supplemental Information 4 Raw data for Figures 1 and 2.

Additional Information and Declarations

Competing Interests

Author Contributions

Animal Ethics

Data Availability

The authors declare that they have no competing interests.

Yinli Guo conceived and designed the experiments, performed the experiments, analyzed the data, prepared figures and/or tables, authored or reviewed drafts of the article, and approved the final draft.

Chengbo Liu conceived and designed the experiments, analyzed the data, prepared figures and/or tables, authored or reviewed drafts of the article, and approved the final draft.

The following information was supplied relating to ethical approvals (i.e., approving body and any reference numbers):

Ethics Committee of Tianjin Jinke Bona Biotechnology Co., LTD.

The following information was supplied regarding data availability:

The raw measurements are available in the Supplemental Files.

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
