# Peer review of "Melatonin attenuates MPP+-induced autophagy via heat shock protein in the Parkinson’s disease mouse model"

_PeerJ, doi:10.7717/peerj.18788_

## Round 0.1 · original submission · Major Revisions

Please provide revisions to the points mentioned by the reviewers. Additionally, revise the abstract for accuracy (it refers to an in vivo model, not in vitro). The methods section of the abstract could be shortened.

The Materials and Methods section also needs revision. The description of the Western blot is poorly written; it lacks details on how proteins were obtained, how much protein was loaded, which gel was used, and what secondary antibody and detection method were employed. For the histology section, the procedure for collecting brain samples needs clarification. It mentions that brain tissues were frozen, but it should also explain how they were embedded in paraffin. The statistical analysis section needs to be revised for accuracy.

Please add a figure showing how the behavioral tests were performed.

The study limitations, such as the very low number of animals per group (n=3), should be discussed. Finally, the Discussion section should be expanded.

Reviewer 1 ·

Basic reporting

The paper investigated the protective effects of melatonin on dopaminergic neurons damaged by MPP+ in a Parkinson's disease model. It finds that melatonin enhances the expression of HSP70, which may inhibit neuronal autophagy and improve motor functions. These results suggest melatonin's potential as a neuroprotective agent in neurodegenerative diseases.
However, there seem to be a number of issues with publishing.
1. There is no clear purpose in the introduction. There is a mention of Parkinson's disease (PD) but no explanation of why the MPTP model was used or what it means.
There is no mention of HSP70, autophagy, CDK5, LC3 II /LC3I, or anything related.
2. Figure 4: The p62 data also seems to show the opposite result, i.e. MPTP treatment should decrease p62, but here it increases.
Reference paper Wang XW, Yuan LJ, Yang Y, Zhang M, Chen WF. IGF-1 inhibits MPTP/MPP+-induced autophagy on dopaminergic neurons through the IGF-1R/PI3K-Akt-mTOR pathway and GPER. Am J Physiol Endocrinol Metab. 2020 Oct 1;319(4):E734-E743. doi: 10.1152/ajpendo.00071.2020.
3. MPTP/MPP(+) has been implicated in neurotoxicity, but there is little experimental evidence for autophagy. References should be provided and mentioned in the discussion.
4. The relationship between autophagy and hsp70 should be referenced and mentioned in the discussion.
5. The Discussion section is too short.

Experimental design

The experimental design is well thought out and well executed.

Validity of the findings

As mentioned in Basic reporting, you need clear results and sufficient narrative in the disscusion section.

·

Basic reporting

Review of the manuscript:” Melatonin attenuates MPP+-induced autophagy via heat shock protein in a Parkinson's disease model” submitted to PeerJ.
Parkinson’s disease is one of the most common and serious neurodegenerative disease, devastating patients and their families both physically and mentally. Drug treatments for Parkinson’s disease can only ameliorate symptoms and do not reverse the disease course. The authors presented new findings pointing to a protective role of melatonin in animal model of Parkinson’s disease. This is an important biomedical area and the results presented in the manuscript will be interesting fir the journal readership. The following corrections should be made.
Abstract. Conclusion. “Melatonin exerts a protective effect against MPP+-induced damage to dopaminergic neurons, potentially by upregulating HSP70 to inhibit neuronal autophagy.” The sense of this sentence is not clear. Presumably, the authors want to say:” Melatonin exerts a protective effect against MPP+-induced damage to dopaminergic neurons, presumably by upregulating HSP70, which inhibits neuronal autophagy.”.
Introduction
“Parkinson's disease (PD) is the second most common neurodegenerative disease among middle-aged and elderly individuals, with an unclear pathogenesis. After this sentence, the authors should add the following reference:”Emamzadeh et al. FN and Surguchov A. Parkinson’s disease: Biomarkers, Treatment, and Risk Factors. Frontiers in Neuroscience, Neurodegeneration, 12, 61230, 2018. https://doi.org/10.3389/fnins.2018.00612
Methods
“After separating the proteins using SDS-PAGE, they were transferred onto PVDF membranes.”
The authors should present the details of the protein transfer to the membrane. Was blotting carried out by electrophoretic transfer? "

Results. Figure 4. The fonts for text on B, C and D are too small, they should be increased for easy seeing. Also, on Fig. 4A CDK5 bands in control and MT look similar, whereas on scanning (B) MT is considerably lower. This needs explanation.
Discussion. “(Rastogi & Haldar.,2020). This study found that melatonin significantly increased HSP70 levels in the PD model, promoting CDK5 expression, reducing LC3B-II and p62 levels, and inhibiting autophagy.” These sentences are confusing, since it might be understood that the finding is related to Rastogi & Haldar.,2020 publication, whereas the authors presumably mean their findings. This should be clearly indicated to avoid confusion.

Experimental design

Experimental design looks good

Validity of the findings

There is no concern about the validity of the findings

Additional comments

The manuscript is well written

---

## Round 0.2 · Minor Revisions

The new version of the manuscript still needs minor revisions for accuracy:

The title should start with a capital letter.

There are still contradictions in the tissue collection description in the methods section. There were only 3 animals per group, so the tissue collection descriptions must be written more clearly throughout the text:

Lines 103-106: “After euthanasia, the hearts were cleaned, and 1/2 fresh specimens of each mouse were fixed with paraformaldehyde and embedded in paraffin for immunohistochemistry. The remaining 1/2 of the substantia nigra tissue was quickly placed in liquid nitrogen or -80°C freezer for WB experiment.” Heart tissue?
Lines 121-122: “The brain tissue was harvested immediately after euthanasia, washed with ice-cold PBS, and then homogenized in ice-cold lysis buffer containing protease and phosphatase inhibitors.” Was it snap-frozen or homogenized?
Lines 147-148: “Mouse brain tissues were harvested for snap-freezing in liquid nitrogen and then stored at -80°C until use or immediately fixed at 4°C for 48 hours with 4% paraformaldehyde (PFA) (pH 7.4).”
Line 117 and Figure Legend 1: Behavior test – According to the description, animals were free-moving in the box. Why was swimming speed recorded? Were the animals in water?
In the figure legend, the word "area," the explanation for the use of color, and the "*" symbol are missing.

Line 155: Please provide the catalog/clone number for the DAT antibody.

·

Basic reporting

The authors corrected the manuscript as recommended by the reviewers and it can be accepted now

Experimental design

OK

Validity of the findings

OK

Additional comments

No problems, the manuscript is improved

---

## Round 0.3 · accepted · Accept

The authors have made the necessary corrections, and the manuscript is now ready to proceed for publication.